# Oxide Free Wire Arc Sprayed Coatings—An Avenue to Enhanced Adhesive Tensile Strength

**Manuel Rodriguez Diaz** [1,*]**, Maik Szafarska** [2] **, René Gustus** [2]**, Kai Möhwald** [1] **and Hans Jürgen Maier** [1]

[1] Institut für Werkstoffkunde (Materials Science), Leibniz Universität Hannover,
30823 Garbsen, Germany; moehwald@iw.uni-hannover.de (K.M.); maier@iw.uni-hannover.de (H.J.M.)

[2] Clausthaler Zentrum für Materialtechnik, Technische Universität Clausthal,
38678 Clausthal Zellerfeld, Germany; maik.szafarska@tu-clausthal.de (M.S.);
rene.gustus@tu-clausthal.de (R.G.)

[*] Correspondence: rodriguez@iw.uni-hannover.de; Tel.: +49-2302-661652

**Abstract:** Conventionally, thermal spraying processes are almost exclusively carried out in an air atmosphere. This results in oxidation of the particles upon thermal spraying, and thus, the interfaces of the splats within the coating are oxidized as well. Hence, a full material bond strength cannot be established. To overcome this issue, a mixture of monosilane and nitrogen was employed in the present study as the atomising and environment gas. With this approach, an oxygen partial pressure corresponding to an extreme-high vacuum was established in the environment and oxide-free coatings could be realized. It is shown that the oxide-free particles have an improved substrate wetting behaviour, which drastically increases the adhesive tensile strength of the wire arc sprayed copper coatings. Moreover, the altered deposition conditions also led to a significant reduction of the coating porosity.

**Keywords:** arc spraying; oxygen-free environment; bonding mechanism

## 1. Introduction

Thermally sprayed coatings are typically applied to enhance certain surface properties of a component, such as its corrosion and wear resistance. In addition to the functional fulfilment, the adhesive tensile strength of the applied coating is crucial [1]. Porosity, residual stresses, degree of oxidation and the prevailing bonding mechanisms all affect the quality of a thermally sprayed coating. The adhesive tensile strength is affected by adhesion and cohesion of the coating. The coating's adhesion is determined by the interaction of physical–chemical mechanisms, such as van der Waals interactions, chemisorption and physisorption, as well as by mechanical interlocking and the wetting behaviour of the impacting particles [2]. In contrast, the coating's cohesion is governed by the binding mechanisms of the splats and the prevailing coating morphology, such as porosity, degree of oxidation, interlamellar fissures, cracks, and residual stresses. Because the initial substrate surface roughening and the actual thermal spraying are both usually carried out in an air atmosphere, oxidation of the substrate surface and the coating particles results. Consequently, metallic and metal-ceramic coatings typically feature heterogeneous coating structures with interlamellar oxide seams and an impairment of the wetting behaviour of the coating layers and substrate surfaces [3]. Hence, the resulting adhesive tensile strength is predominantly determined by mechanical interlocking effects. Accordingly, the bond strengths of non-ferrous arc-sprayed coatings is typically in the range of only 14–41 MPa [4,5]. Furthermore, technological properties such as wear resistance, corrosion resistance, and machinability of metallic coatings can be negatively affected by oxidation of the alloying elements [6–8].

These challenges are well known and some technical solutions have already been established. One commercially realized approach is cold gas spraying (CGS). The CGS

process is characterized by coating particles that only experience temperatures in the process that are significantly below the respective melting temperature. Due to the low temperatures employed in combination with the prevailing high kinetic energy of the particles, very dense coatings with low oxide content can be realized [9–11]. Yet few coating materials provide sufficient ductility to be used for CGS. Consequently, ceramics and metal carbides can only be applied with great effort and to a very limited extent [12]. Another main disadvantage arises due to the plastic deformation process, which results in small residual ductility of the final coating [13–15]. In addition, the high impact velocities of the particles induce relatively high residual stresses in the coatings and the substrates, which makes coating of thin-walled components challenging [16,17]. Thus, CGS has only reached small market penetration [18,19].

Another technical solution is to transfer the thermal spraying processes to a vacuum or an inert gas atmosphere (argon or nitrogen, depending on the coating material) [18,20,21]. However, only a rough vacuum ($p$ > 5 kPA) can typically be established in the process, as plasma and powder conveying gases are needed with this technology [21]. It is reported that a NiCoCrAl coating applied under such conditions contained 675 ppm oxygen, and a nickel–titanium coating contained 1086 ppm oxygen [22]. In addition to vacuum plasma spraying, radio frequency inductively coupled plasma spraying (RF-ICP) or vacuum induction plasma spraying (VIPS) provides the possibility of producing coatings in a controlled process atmosphere (vacuum or inert gas) [23]. Samal et al. reported on oxide-free NiTi coatings produced with this method [24]. However, as in vacuum plasma spraying, the oxygen content in this process is determined by the adjustable vacuum and the residual oxygen content of the used commercial inert gases.

As an alternative, the free jet can be shielded from the surrounding air atmosphere by a coaxially enveloping protective gas shield. Usually, a ring gas nozzle is attached to the spray gun [7,25,26], and a reduction in the oxide content and porosity of the coating can be realized. To ensure that process gases and overspray can flow out, a spray gap must be kept open between the shroud and the substrate surface, and thus, oxidation cannot be completely prevented [25,27]. Furthermore, the splat surfaces are exposed to the ambient atmosphere after each coating overrun, resulting in passivation of the splat surfaces. Consequently, the adhesive tensile strengths of shrouded coatings is not increased [28,29].

Although the previous processes can noticeably reduce oxide formation in thermally sprayed coatings, oxidation and passivation of the interfaces and surfaces cannot be completely avoided. Consequently, a metallurgical bonding of the spray particles, and thus, a full material-locking wetting, is not possible. Recent investigations have demonstrated that silane doping of an inert process gas atmosphere can completely eliminate the residual oxygen and water content at room temperature, and results in complete flux-free wetting of the component surface by a molten metal [30]. Recently, this approach was exploited to realize the wire arc spraying process in an oxygen-free atmosphere [31]. In the present study, the resulting coating morphologies and properties using this approach are presented for pure copper coatings.

## 2. Materials and Methods

### 2.1. Enviromental Conditions

To realize an oxygen-free spraying process, a 0.26 m$^3$ coating chamber with a manipulation device and a lambda probe for oxygen monitoring was designed, Figure 1.

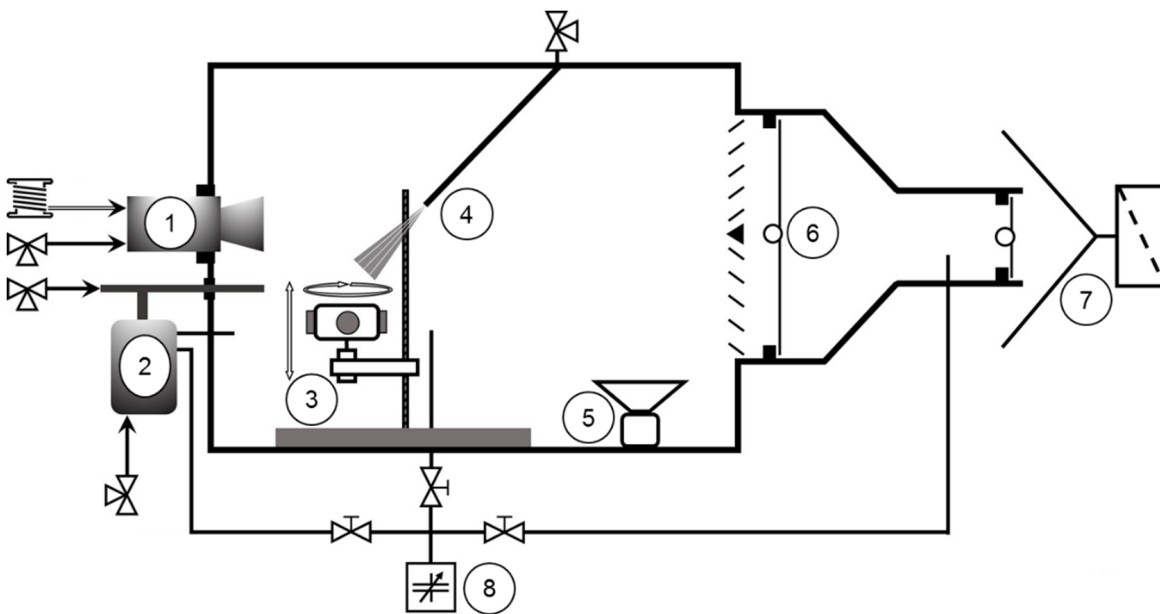

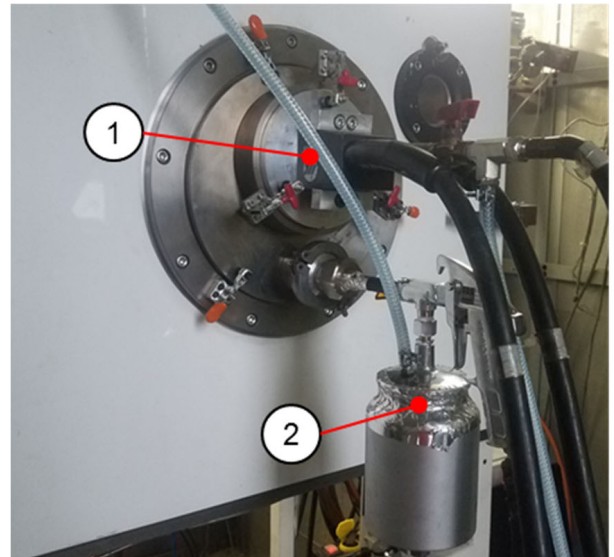
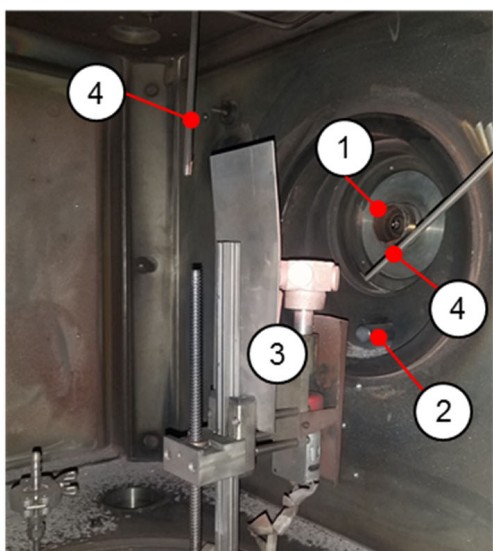

(1) wire arc spraying system
(2) corundum blasting system
(3) manipulation device
(4) cooling and flushing system

(5) particle collector
(6) check valve system
(7) extraction and filter system
(8) lambda probe

**Figure 1.** Schematic of the setup used for arc spraying in an oxygen-free environment.

In this setup, both the free jet and the substrate are fully contained with the oxygen-free atmosphere. Furthermore, a blasting device can be flange-mounted such that the activation of the substrate surface can be realized in the oxygen-free environment as well.

To achieve an oxygen-free environment in the chamber, a nitrogen atmosphere of commercial quality with ambient pressure was established first. The residual oxygen and residual water contents present in such an atmosphere are typically around 20 ppm. Both the residual oxygen and the water contents were then quantitatively converted to silicon dioxide ($SiO_2$) and hydrogen ($H_2$) by adding monosilane ($SiH_4$) with the stoichiometric ratio of the conversion reaction:

$$SiH_4\ (g) + O_2\ (g) \rightleftharpoons SiO_2\ (s) + 2H_2\ (g) \tag{1}$$

$$SiH_4 \text{ (g)} + 2H_2O \text{ (g)} \rightleftharpoons SiO_2 \text{ (s)} + 4H_2 \text{ (g)} \qquad (2)$$

The quantities of silicon dioxide obtained per second ($q_s$) for a given protective gas flow ($Q_{Sg}$) with the oxygen ($X$) and water ($Y$) concentrations to be eliminated were obtained as:

$$q_s = 0.0025 \times Q_{Sg} \, (X + 0.5 \times Y) \qquad (3)$$

Thus, in an inert gas atmosphere with 20 ppm residual oxygen and a total protective gas flow of 120 m³/h during the coating process, approx. 7.5 g/h of silicon dioxide are produced. As shown in [30,31], the silicon dioxide formed is amorphous, which is nontoxic. From a safety point of view, it is also important to note that pre-diluted (2 vol.-%) monosilane in an inert gas was used to create the oxygen-free environment. Such low concentrations do not generate inadvertently flammable gas mixtures.

However, to ascertain the absence of oxygen during the entire coating process, setting the process atmosphere alone is not sufficient. Thus, both the atomisation process and the blasting process were also carried out with silane-doped nitrogen. For this purpose, purge and bypass lines were connected to the spraying and blasting system. The oxygen content in the environment during the blasting and coating processes was $\approx 1 \times 10^{-26}$ vol.-%, which corresponds to an oxygen content of $\approx 1 \times 10^{-23}$ Pa, cf. Figure 2. With this approach, the residual oxygen contents of the process and ambient gases during the whole coating process corresponded to those present in an extreme-high vacuum (XHV) [32].

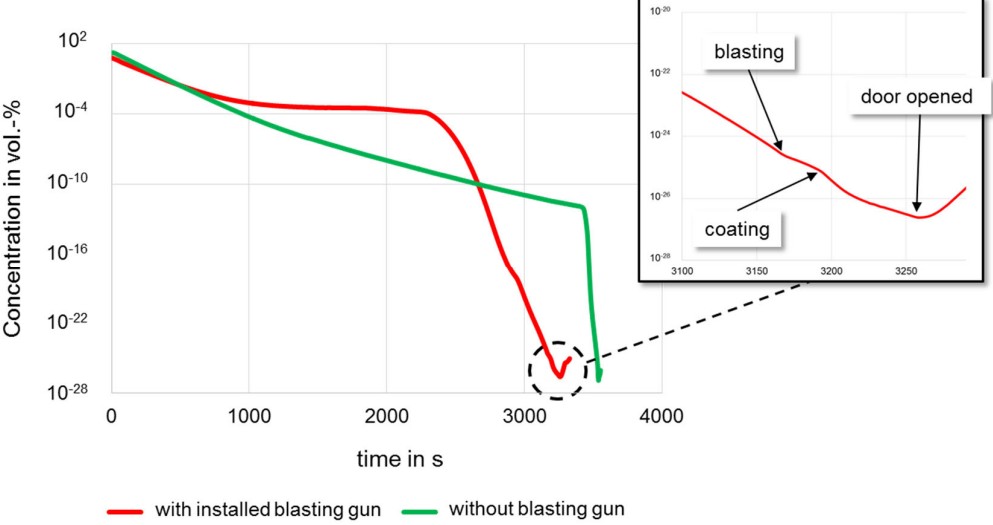

**Figure 2.** Evolution of the oxygen concentration as monitored with a lambda probe during flushing, blasting, and the actual coating process.

The curves depict the decrease of oxygen concentration within the process chamber over time with an attached blasting gun installed (red) and without a blasting gun (green). As can be seen from the curves, the oxygen concentration decreased slower with the installed blast gun than without. The reason for this is the corundum tank. Due to the micro-cavities between the corundum particles, it is more time consuming to create a pure nitrogen-based atmosphere. However, in order to keep the process time within acceptable limits with the blasting gun installed, the addition of monosilane was started at an earlier stage (t ≈ 2200 s), with a high residual oxygen content ($\approx 1 \times 10^{-4}$ vol.-%). In contrast, the oxygen concentration without the installed blasting gun decreased almost linearly to $1 \times 10^{-12}$ vol.-% within 3300 s. From this oxygen concentration onwards, the residual oxygen content of the environment was further reduced by the addition of monosilane.

## 2.2. Coating System and Materials

Coating tests were carried out both in air atmosphere and in the XHV-adequate environment. All coatings presented in this study were carried out with the same coating parameters. The coatings were carried out with a wire arc spraying system, Miller BP400 (Modularc 400R, Miller Thermal, Inc., Appleton, WI, USA). S235JR (1.0038) steel discs with a diameter of 25 mm and a thickness of 5 mm were used as substrate. Copper wire with 1.6 mm in diameter was employed as coating (GTV 50.12.). The spray distance was 100 mm and 130 g of the copper wire was atomised per minute. Furthermore the voltage and amperage were 30 V and 100 A.

To study the effect of the initial surface condition of the substrate, the substrate activation by corundum blasting (EKF 36, 800 kPa) was conducted both in conventional air and in the process chamber under oxygen-free conditions. The aim of activating the substrate surface in the oxygen-free environment was to create an oxide-free interface between the substrate and the coating to improve wetting behaviour, and thus, coating adhesion. Conventional substrate surface activation was carried out in an injector blasting cabin (Goldmann Spezial 1000) (Friedrich Goldmann GmbH & Co. KG, Mannheim, Germany). For oxygen-free corundum blasting, a modified sandblasting gun (Schneider SSP-SAV) (Schneider Druckluft GmbH, Reutlingen, Germany) was used. The modification included the supply device for silane into the corundum carrier gas (nitrogen). Furthermore, the corundum tank was modified to allow flushing with nitrogen and flooding with silane-doped nitrogen, thus creating an oxygen-free environment in the tank. In addition, the corundum tank was equipped with a bypass line to allow oxygen concentration monitoring with a lambda probe (MESA Mess- und Regeltechnik, Filderstadt-Bonlanden, Germany).

## 2.3. Characterization of the Coatings

The morphology of the coatings was analysed using reflected light microscopy. Additionally, scanning electron microscopy (SEM) of the cross sections were performed using a Helios Nanolab 600 (FEI Germany GmbH, Frankfurt, Germany) with a dual-beam system operating under high vacuum conditions with a base pressure of $10^{-4}$ Pa. The porosity was determined by digital analysis using the software ImageJ (1.50i, Wayne Rasband (National Insitutes of Health, Bethesda, MD, USA) [33]. The surface roughness of the coatings was determined with a surface roughness tester Mitutoyo Surftest SJ-210 (Mitutoyo Europe GmbH, Neuss, Germany). A diamond tip of 2 $\mu$m/60° was used at a traverse speed of 0.75 mm/s and a measuring force of 4 mN.

In order to study the effect of the different environmental conditions, adhesion tensile strength tests were carried out in accordance with the standard DIN EN ISO 14916, sample type B, cf. Figure 3. In these tests, Ultrabond 100 was used as the adhesive.

Table 1 gives an overview of the sample sets codes in this study. Sample sets I–V were coated with the parameters presented in Section 2.2; only the coating and blasting environment was varied.

**Table 1.** Sample set codes of the adhesive tensile strength specimens.

| Coating Specimens in Accordance to DIN EN ISO 14916 | | |
|---|---|---|
| Sample set | Coating environment | Blasting environment |
| I | air | air |
| II | silane + $N_2$ | air |
| III | silane + $N_2$ | silane + $N_2$ |
| IV | silane + $N_2$ | silane + $N_2$ |
| V | silane + $N_2$ | silane + $N_2$ |
| | | Result validation specimens |
| Sample set | Description | |
| VI | 25 × 25 × 0.5 mm pure copper sheets | |
| VII | adhesive blank test | |

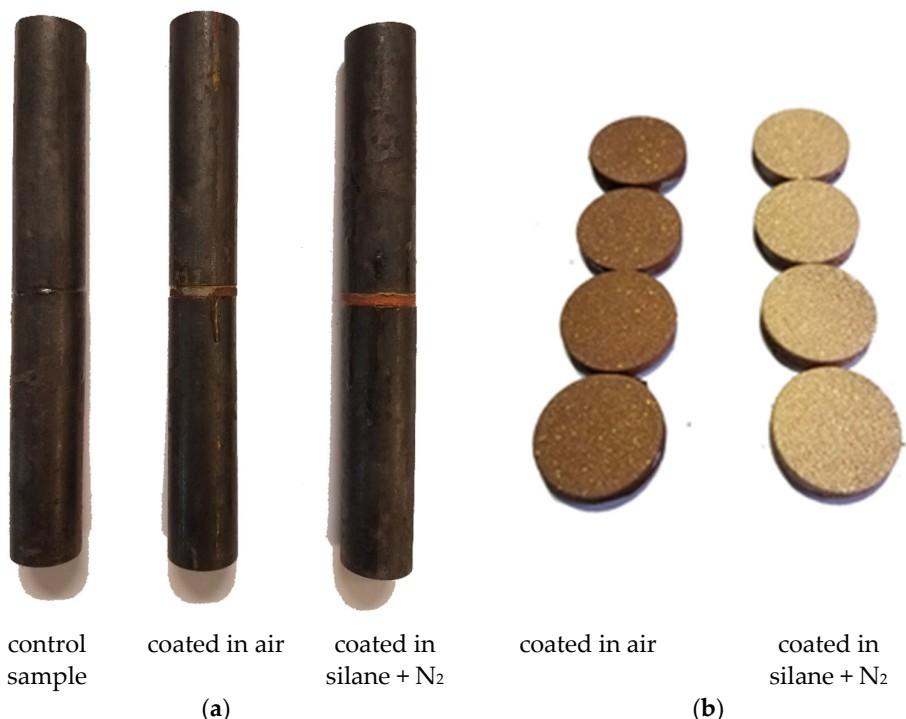

| control sample | coated in air | coated in silane + N₂ | coated in air | coated in silane + N₂ |

(**a**)  (**b**)

**Figure 3.** Adhesive tensile strength specimens in accordance to DIN EN ISO 14916: (**a**) used sample type B setup; (**b**) 25 mm diameter S235JR discs coated with copper in air and in an atmosphere with an oxygen content of less than $1 \times 10^{-26}$ vol.-%.

In order to make statements about reproducibility, the tests, according to the sample set III, were repeated twice (sample sets IV and V). The results obtained by these reproducibility tests led to additional tests (result validation specimens VI and VII). Sample set VI contained five $25 \times 25 \times 0.5$ mm copper sheets, bonded between two steel stamps with the aim to validate the adhesive limits on pure copper. Sample set VII was a blank sample, where the steel stamps were directly bonded together in order to test the properties of Ultrabond 100.

## 3. Results

### 3.1. Coating Morphology

In Figure 4, the coating morphologies of copper coatings formed in an air atmosphere and in an oxygen-free environment are compared. The difference with respect to porosity and oxide content are striking.

The copper coatings produced in the oxygen-free environment were free of oxide fringes. Furthermore, the gaps at the interface between the substrate surface and the coating surface were significantly reduced in this case, and few interlamellar gaps were present. Moreover, the coating porosity was noticeably reduced from 17.5% (Figure 4b) to 5.3% (Figure 4d).

Whereas the actual coating was performed in the XHV-adequate environment, blasting in this case was conducted in conventional air. Thus, the substrate surface featured an oxide layer, which was expected to have a negative effect on wetting. Thus, the corundum blasting process was also conducted within the XHV-adequate environment in the subsequent tests. Figure 5 shows SEM images from the substrate/coating interfaces that reveal the substantial effect obtained upon transferring the corundum blasting process to XHV-adequate environment.

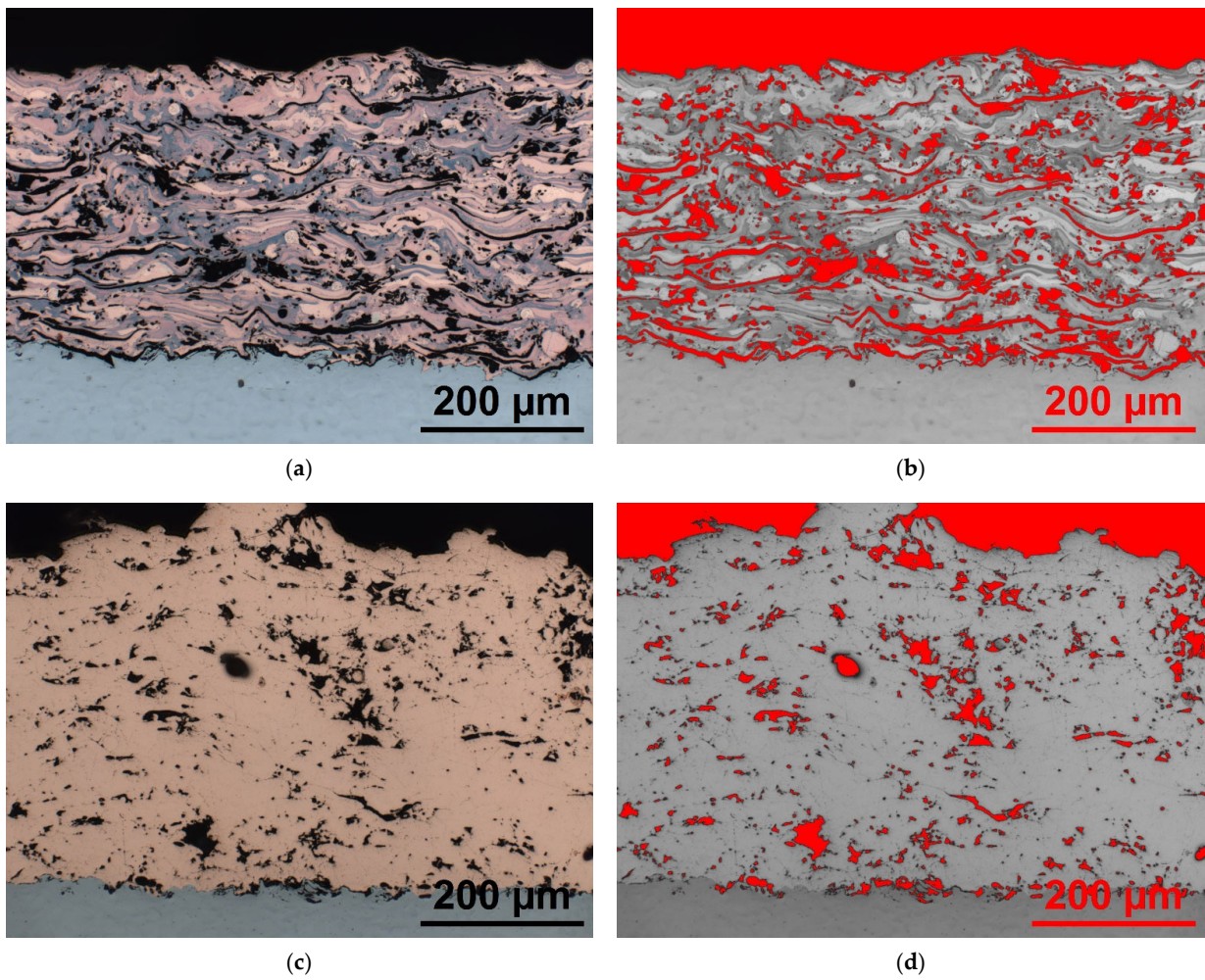

**Figure 4.** Reflected light microscopy images obtained on cross sections of: (**a**) conventionally in air sprayed wire arc copper coating with (**b**) porosity analysis and (**c**) copper coating created in an environment with an oxygen content of $1 \times 10^{-26}$ vol.-% with (**d**) porosity analysis.

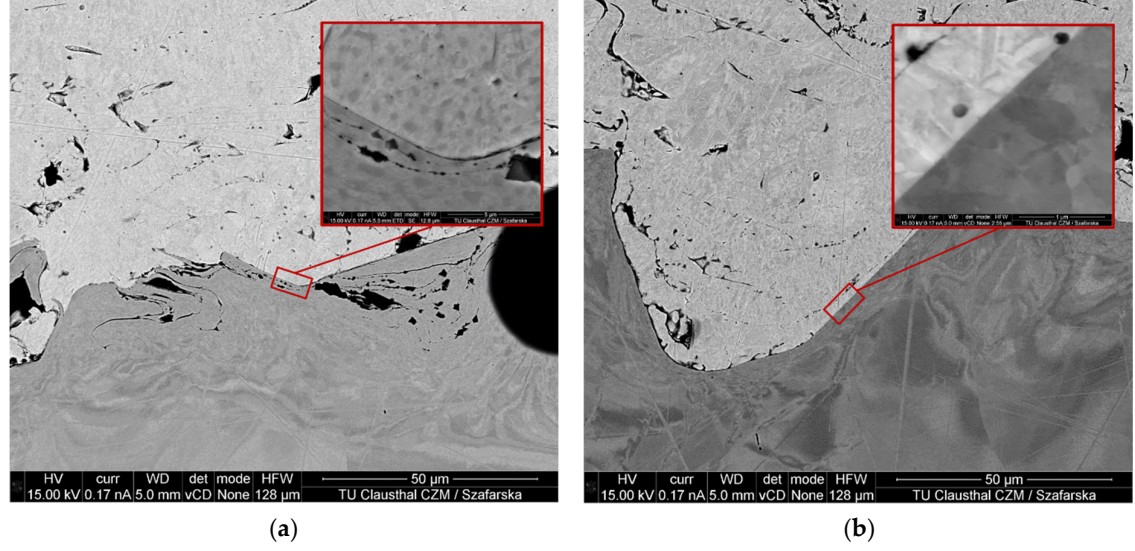

**Figure 5.** SEM micrographs of cross sections from samples coated with copper in silane-doped nitrogen, which were blasted: (**a**) in air and (**b**) in the XHV-adequate environment.

Figure 5a shows an oxide-free layer where the substrate surface was conventionally activated by blasting in air. Despite improved wetting behaviour, a small but clear gap between the coating and the substrate is seen along the entire interface. By transferring the blasting process to the oxygen-free environment, the wetting-inhibiting oxide film can be completely removed from the surface. As no oxygen was present in the process atmosphere, no new wetting-inhibiting oxide film could form after blasting, and thus conditions for complete wetting of the substrate surface were realized. In fact, as shown in Figure 5b, regions are now seen where no interfacial gap was detectable in the SEM micrographs.

To quantitatively highlight the oxidised phases, EDS mappings were performed (Figure 6) by splitting the SEM picture into 256 × 224 individual EDS spectra, each represented by one pixel (57,344 in total). Afterwards, the amount of pixels were halved for a better visual representation. All spectra were taken with an acceleration voltage of 15 kV and 0.34 nA current.

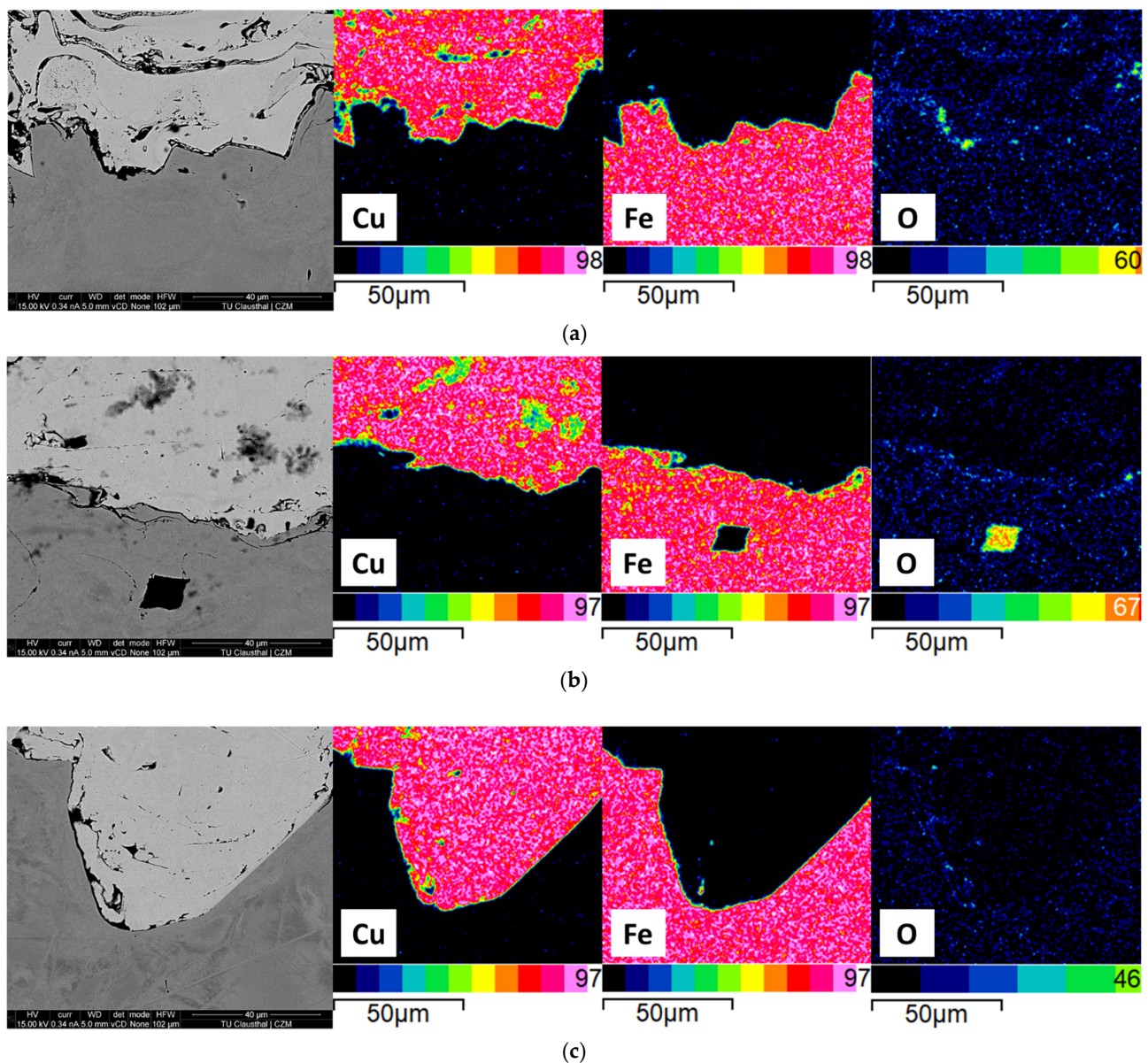

**Figure 6.** SEM micrographs and quantitative EDS mappings of cross sections from copper coatings, which were (**a**) blasted and coated in air, (**b**) blasted in air, coated in the XHV-adequate environment, and (**c**) blasted and coated in the XHV-adequate environment. EDS scales in atomic percent.

On the one hand, the significantly lower oxygen content within the copper coatings that were coated in a silane-doped nitrogen environment was clearly visible. The fact that the coatings do not appear completely oxygen-free was due to the coating porosity and oxidation processes during sample preparation. Furthermore, oxygen-free interfacial transitions areas were shown by blasting under XHV-adequate conditions (Figure 6c).

### 3.2. Adhesion Tensile Strength

In the following, the effects of the different wetting behaviours discussed in the previous chapter are related to the resulting failure mechanisms. As shown in Figure 7, both the adhesive tensile strength values and the corresponding fracture appearance are substantially affected by the environmental conditions present upon blasting and coating.

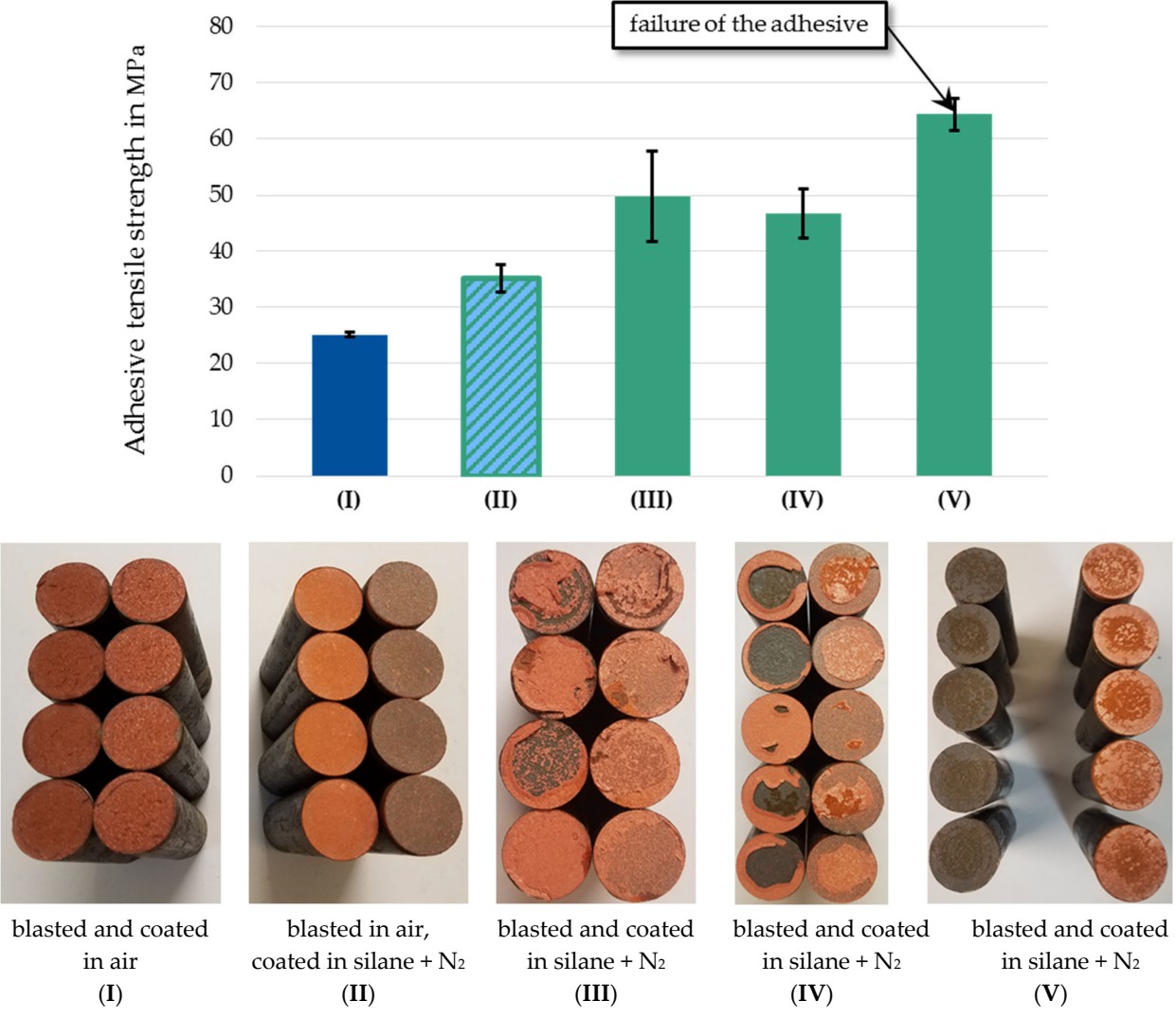

**Figure 7.** Adhesive tensile strengths and fracture surfaces of wire arc sprayed copper coatings upon blasting and coating in different environments.

Specimen set (I), which was coated and blasted conventionally in an air environment using compressed air as atomising and blasting gas, represents the standard reference coating. In this case, the adhesive tensile strength was $25.1 \pm 0.4$ MPa. All these samples showed exclusively cohesive coating failure. Sample set (II) was blasted conventionally in an air environment but coated using silane-doped nitrogen as environment and atomising gas. By transferring the coating process to an oxygen-free process environment, the adhesive tensile strengths could be substantially increased, and sample set (II) reached

35.0 ± 2.4 MPa. This value corresponds to that by Gourlaouen et al. who presented adhesive tensile strengths of wire arc copper coatings produced in nitrogen [34]. In this set, all samples showed adhesive coating failure, which can also be noted from the difference in colour of the respective fracture surfaces. Specimen sets (III), (IV), and (V) were all blasted and coated in silane-doped nitrogen. The adhesive tensile strengths of the sample set (III) were 49.8 ± 8.8 MPa, which corresponds to an increase of 98% compared to the sample set (I). The experiments were duplicated twice and sample set (IV) demonstrated an 87% increase in adhesive tensile strength of 46.8 ± 4.4 MPa, whereas the highest adhesive tensile strengths were achieved by sample set (V). In this case, the adhesive tensile strength was 63.9 ± 3.0 MPa, which is an increase of 154% compared to sample set (I). Whereas sample sets (III) and (IV) showed mixed fractures, sample set (V) showed solely the failure of the adhesive. The reasons for the larger variation in adhesive tensile observed for the samples blasted and coated in the XHV-adequate environment will be discussed in Section 4.2.

## 4. Discussion

### 4.1. Coating Morphology Depending on the Setted Process Atmosphere

The previous results demonstrated that processing under oxygen-free conditions has a substantial influence on the resulting coating morphology. In fact, conditions could be created that allow the splats to form a material bond with each other upon deposition. Blasting under oxygen-free conditions also improves the wetting behaviour on the substrate surface to such an extent that interfacial gaps are minimised and partially or even completely closed. Although the improved wetting behaviour had a reducing effect on the resulting coating porosity, the coatings were not free of pores and are still far from being able to meet the low porosities of, for example, cold gas sprayed copper coatings. In order to gain a more detailed understanding of pore formation, the particle formation under oxygen-free conditions was studied. As it was not possible to integrate a classic particle diagnostics system into the designed chamber, the splat formation was characterized for the different environmental conditions employed. Figure 8 shows representative examples of the splats formed on the substrate surface. In all cases, the splats formed in the XHV-adequate environment were substantially different in shape from the conventional ones.

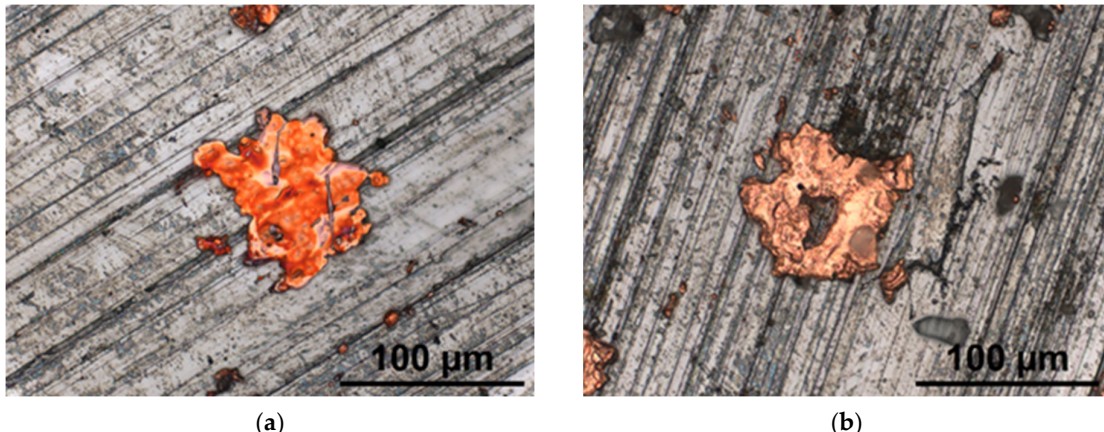

(**a**)                              (**b**)

**Figure 8.** Reflected light microscopy images of wire arc copper splats formed upon impact on S235JR samples: (**a**) conventional spraying in air and (**b**) spraying in silane-doped nitrogen.

For actual comparison, the focus was placed on splats of similar size because it can be assumed that the particles then also have similar kinetic and thermal energy due to the similar properties of air and nitrogen. In this case, the different formation of the splats was then essentially controlled by the oxide film-dependent surface tension of the impacting particles. While the splats formed with air as atomising and atmospheric gas have the typical pancake shape, the oxide-free particles are clearly distinguished by

their dispersion behaviour on the substrate surface. The oxide-free formed splats had a doughnut-like shape and were more frayed around the edges than the conventional splats. This doughnut-shaped spreading behaviour appears to be one of the main reasons why the wetting behaviour was impaired despite the absence of oxygen. Thus, the oxide-free formed coatings have a significantly lower, but still unexpectedly high, porosity.

However, because this alone cannot be the only reason for the remaining coating porosity, the particle formation itself was investigated. Particle formation upon atomisation is also dependent on the environmental conditions and can be described with the Weber number [35–38]:

$$W_e = (\rho_g \times v^2 \times D_p) / \sigma \tag{4}$$

where $\rho_g$ is the atomization gas density, $D_p$ is the droplet diameter, $v$ is the atomization gas velocity, and $\sigma$ is the surface tension. The key parameter that was varied upon moving from a process in air to a process in an XHV-adequate environment was surface tension of the particles, as this is reduced if no oxide layer is present [34,39]. In turn, the Weber number for similar free jet conditions was higher in the XHV-adequate environment, which resulted in a more drastic particle break up. This effect was clearly reflected in SEM images obtained from cross sections of coatings formed in the XHV-adequate environment, cf. Figure 9.

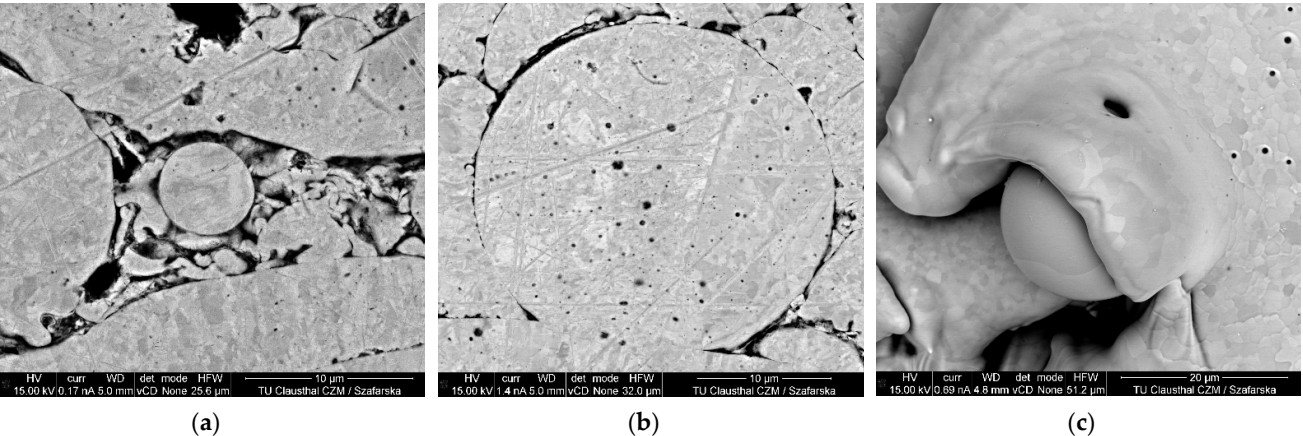

(**a**) (**b**) (**c**)

**Figure 9.** SEM micrographs of undeformed and trapped copper particles in wire arc copper coatings formed in an XHV-adequate environment: (**a**) particle entrapment in the vicinity of pores, (**b**) partially bonded copper particle and (**c**) by splat partially covered particle on the coating surface.

In contrast to conventional coatings, oxide-free coatings contained a large number of very small undeformed particles in the immediate vicinity of pores (Figure 9a). Although the undeformed particles were able to form a material bond locally in the impact area due to the lack of a wetting-inhibiting oxide film (Figure 9b), the process-dependent kinetic energy levels during impact were too low to completely suppress pore formation (Figure 9c).

These very small particles could be detected not only in the areas near pores, but also on the coating surface. Figure 10 presents a comparison of coating surfaces formed in air and in silane-doped nitrogen. The copper coatings produced in air (Figure 10a) feature the expected surface morphology [40]. In contrast, the surface appearance of the coatings produced in silane-doped nitrogen is substantially different (Figure 10b). Clearly, the coating surface had a large number of very small particles due to the effect of the reduced surface tension upon atomisation. Still, the oxide-free coating surface appears smoother after particle impact, as shown in Figure 11.

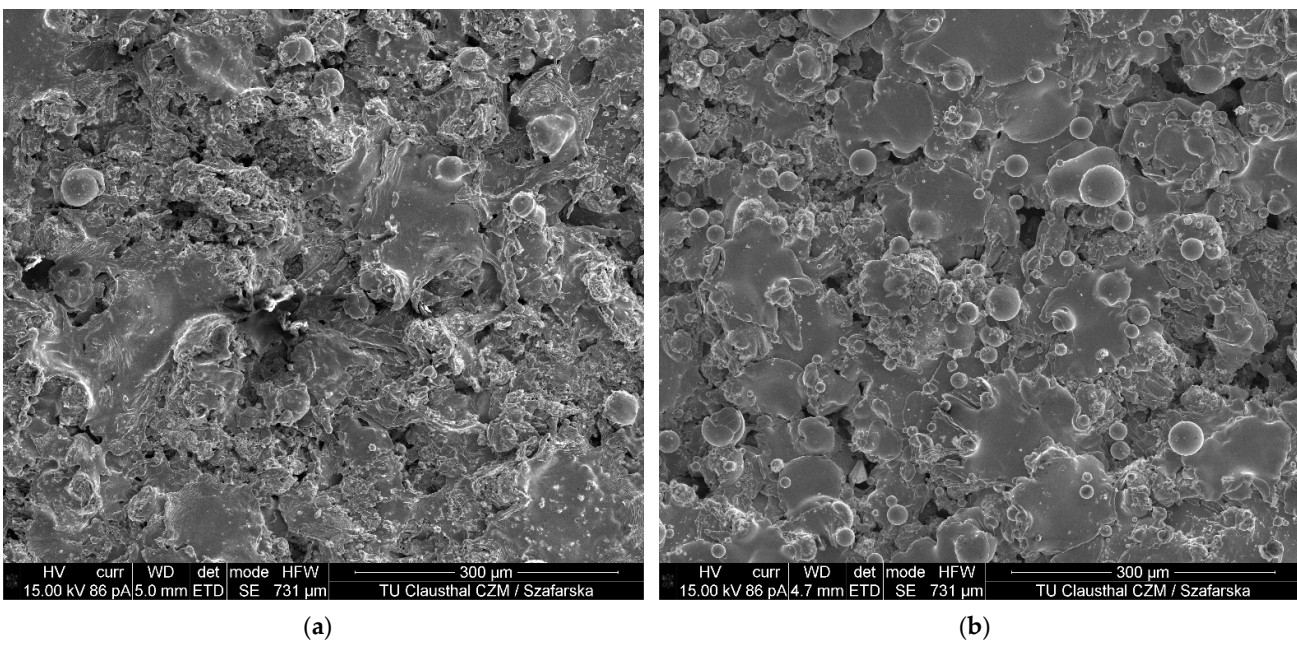

**Figure 10.** SEM micrographs of the surfaces of wire arc copper coatings: (**a**) conventionally coated in air; (**b**) coated in silane-doped nitrogen.

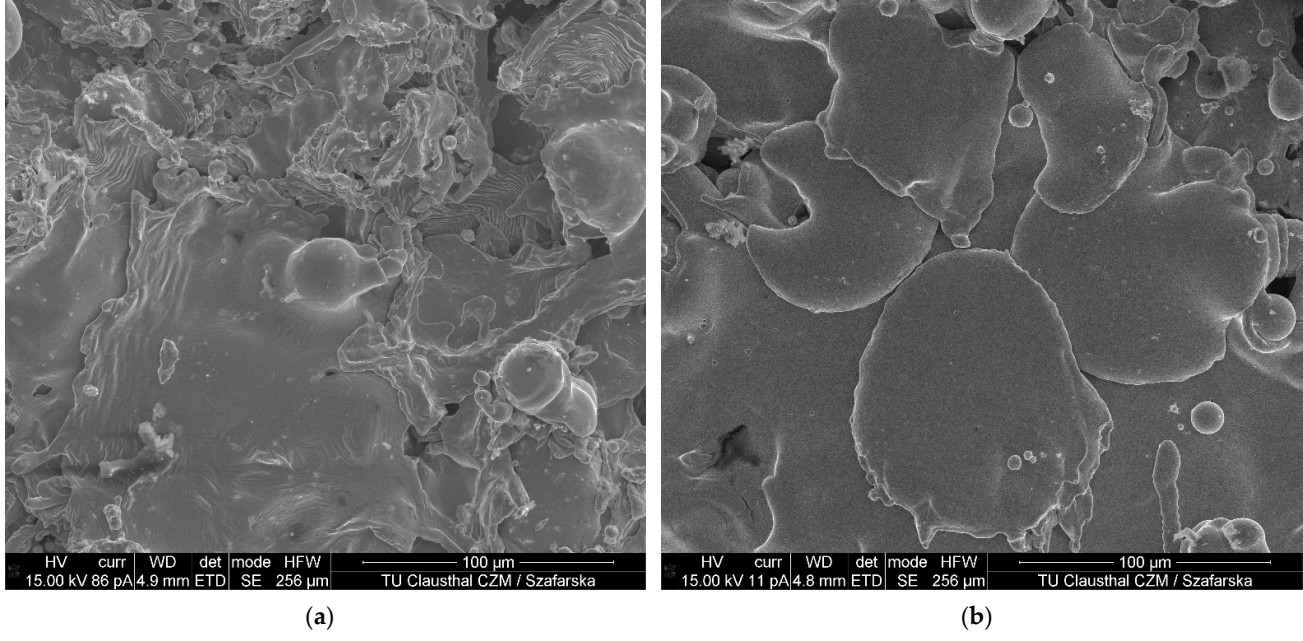

**Figure 11.** SEM micrographs of wire arc copper coating splats on coating surface: (**a**) conventionally coated in air; (**b**) coated in silane-doped nitrogen.

Although the oxide-free splats have a smoother appearance, the small particles on the coating surface provide a slightly rougher coating surface, as shown by the surface roughness analysis in Figure 12.

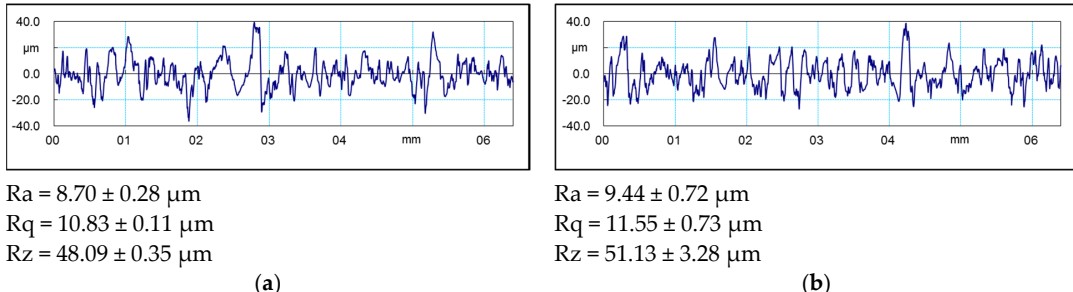

Ra = 8.70 ± 0.28 μm
Rq = 10.83 ± 0.11 μm
Rz = 48.09 ± 0.35 μm

(**a**)

Ra = 9.44 ± 0.72 μm
Rq = 11.55 ± 0.73 μm
Rz = 51.13 ± 3.28 μm

(**b**)

**Figure 12.** Roughness evaluation profiles and roughness of wire arc copper coating surfaces: (**a**) conventionally coated in air; (**b**) coated in silane-doped nitrogen.

Another difference can be seen when comparing the individual particles captured with a particle collector installed in the chamber. As shown in Figure 13, conventional particles are slightly oval-shaped and their surface is rugged (Figure 13a,b). In contrast, oxide-free formed particles deviate less from a round shape and have a significantly more homogeneous and smoother surface (Figure 13c,d). Again, the effect is attributed to the reduction in surface tension of the particles in the oxygen-free environment. A valid particle size distribution could not be determined. Due to the high turbulence and the small coating volume inside the process chamber, it has to be assumed that a representative collection of the smallest particles was not obtained. These particles have a too low mass and are mostly carried out of the chamber with the overspray.

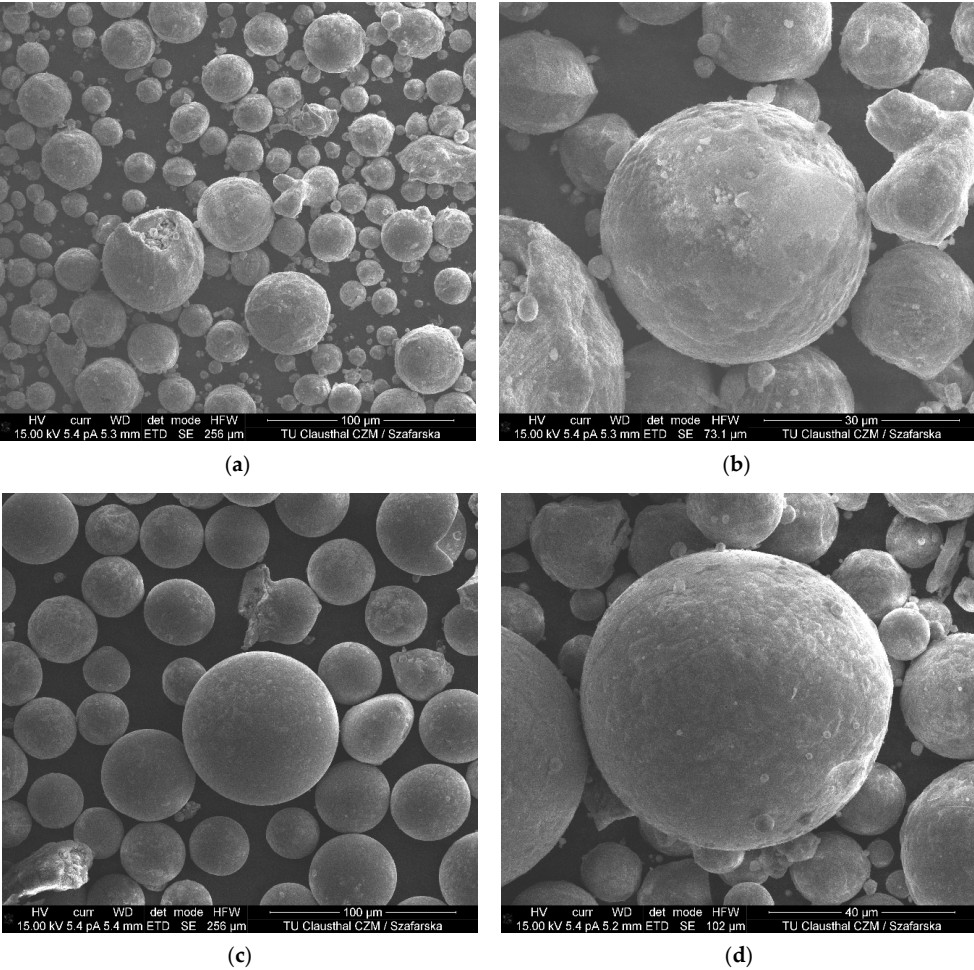

(**a**)

(**b**)

(**c**)

(**d**)

**Figure 13.** SEM micrographs of wire arc copper particles: (**a**,**b**) formed upon spraying in air and (**c**,**d**) in silane-doped nitrogen formed.

### 4.2. Adhesive Tensile Strength

The comparison of the data shown in Figure 7 for sample set (I) with the coating morphology presented in Figure 4a reveals that the oxidation of the particles during the flight phase is the limiting factor for bond strength. Based on the fracture analysis, the interfaces and interlamellar gaps are the weak points of a conventionally sprayed coating.

In contrast, all fracture surfaces of sample set (II) showed pure adhesive coating failure. This demonstrates that the oxide-free interfaces between the individual splats resulted in a material bonding between them, which increased the cohesion within the coating. However, due to the conventional blasting in air atmosphere, there was still an oxide-containing interface between the actual substrate and the coating. This limits the wetting, and thus the adhesion of the coating. Consequently, the bonding between the substrate and the coating is governed mainly by mechanical interlocking, and thus represents the limiting factor of the sample set (II).

By transferring the blasting process to the oxygen-free environment as well, the adhesive tensile strengths could be significantly increased up to the point of failure of the adhesive in the case of the sample set (V). In order to understand the reasons for the larger scatter seen in Figure 7 for the samples sets (III), (IV), and (V), additional tests were carried out with drawn copper sheets and pure substrate material. The data obtained are summarized in Figure 14 along with the data for sample set (V).

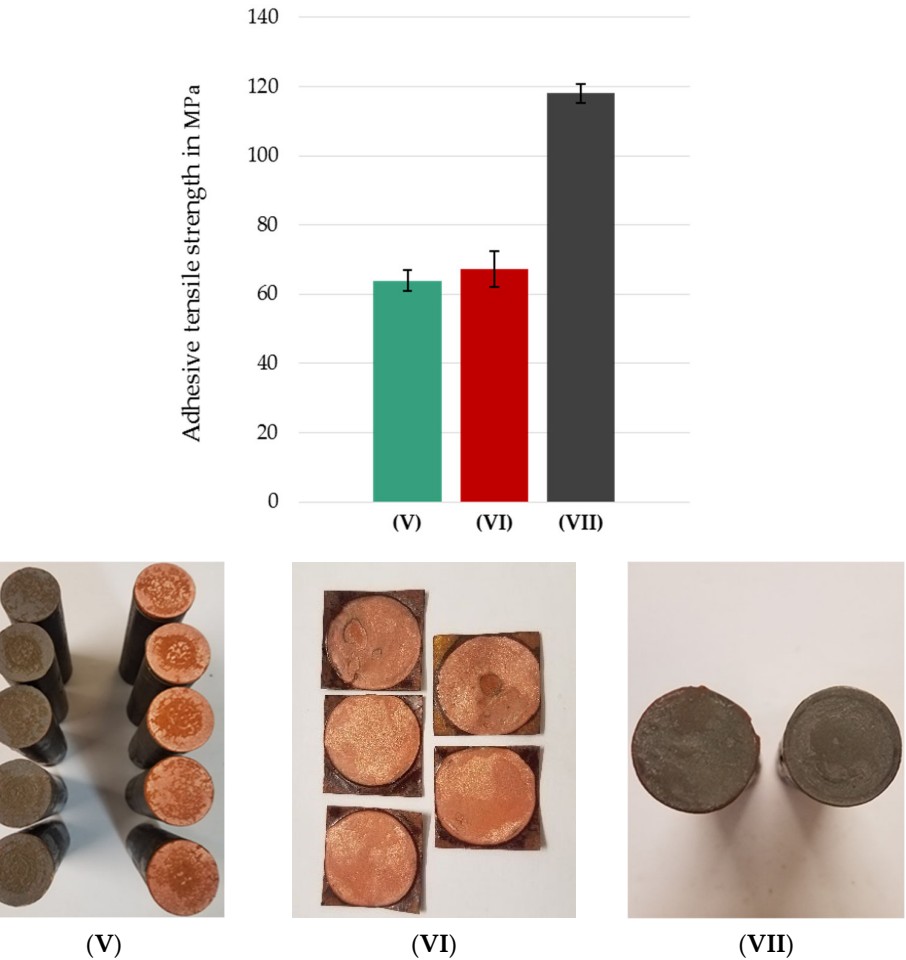

**Figure 14.** Adhesive tensile strength and fracture surfaces of (**V**) samples blasted and coated in silane-doped nitrogen, (**VI**) pure copper sheets, and a (**VII**) control sample from sample set (**V**).

Ultrabond 100 is known to feature a decrease in adhesive properties over time [41]. Thus, a control sample was produced in which the substrates were directly bonded to each

other, i.e., without any prior coating. The control sample (Figure 14VII) demonstrated an adhesive tensile strength of 114.7 MPa. Accordingly, the adhesive Ultrabond 100 achieved the specified minimum adhesive tensile strength of 100 MPa. Next, pure copper sheets with a thickness of 0.5 mm, which roughly corresponds to the thickness of the thermally sprayed coatings, were tested, cf. Figure 14VI. In this case, the adhesive failed at $67.3 \pm 5.1$ MPa. This proves accordingly to [42] that Ultrabond 100 has a reduced adhesion to copper, which in turn limited the strength values that could be determined for the coatings formed in silane-doped nitrogen. This indicates that the true maximum adhesive tensile strength of these samples sets might be even higher. Yet it does not explain the larger scatter in the data seen for sample sets (III), (IV), and (V), cf. Figure 7, despite their nominal identical processing. However, a detailed analysis of the interface between the substrate and the coating revealed that corundum blasting in the chamber itself is not an optimal solution for surface activation. Although the oxide film can be removed by blasting in the silane-doped nitrogen environment, corundum particles can be trapped at the interface between substrate and coating, cf. Figure 15. These particles do act as local stress raisers that ease crack nucleation. Depending on the shape, size, their local density, and orientation, the resulting adhesive tensile strength is more or less affected. Due to the design of the current process, oxygen-free cleaning of the substrate surface after blasting is not yet possible.

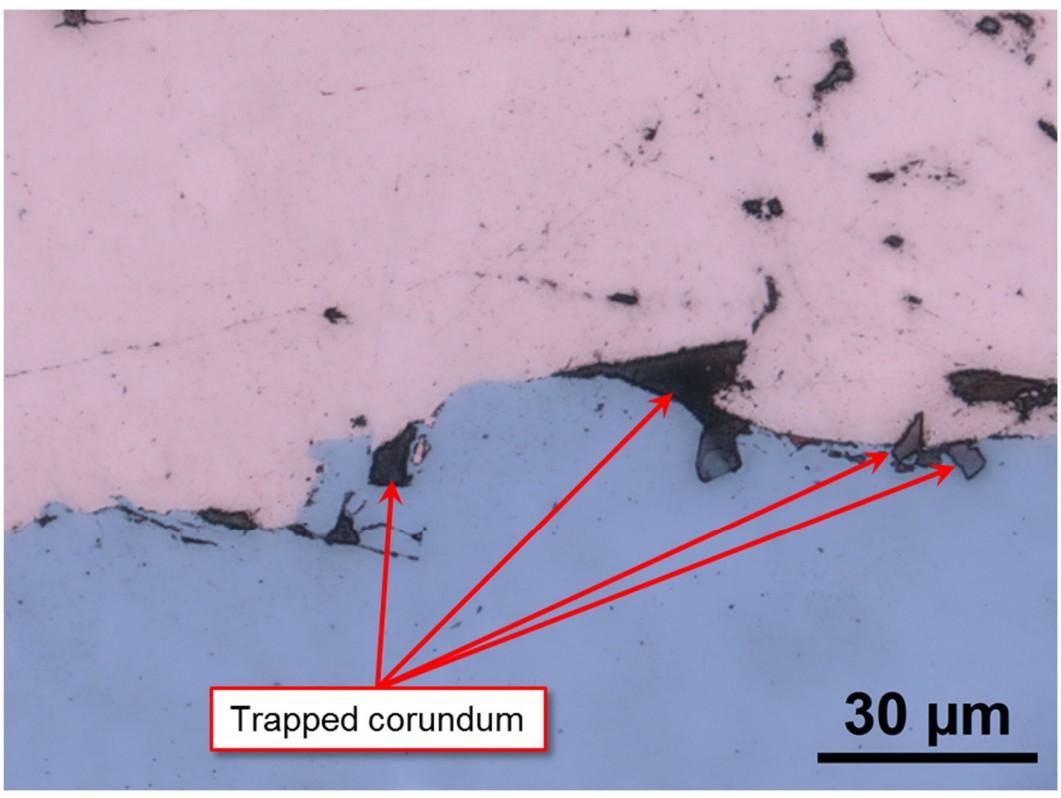

**Figure 15.** Reflected light microscopy image of a cross section of a copper coating, blasted and coated in the chamber, revealing trapped corundum particles at the interface.

Another factor is the installed sandblasting gun itself. As described in Section 2.1, it is more time consuming to realize an oxygen-free environment due to the corundum tank. To reduce overall processing time, the corundum tank was flushed with silane-doped nitrogen at an early stage, despite the still relatively high residual oxygen content. As a result, a substantial amount of silicon dioxide was produced in the corundum tank. As shown in Figure 16, the fine silicon dioxide is carried on into the coating chamber during the blasting process. Part of these remain in the atmosphere and are later deposited on the copper particles and the specimen surface during the actual coating process.

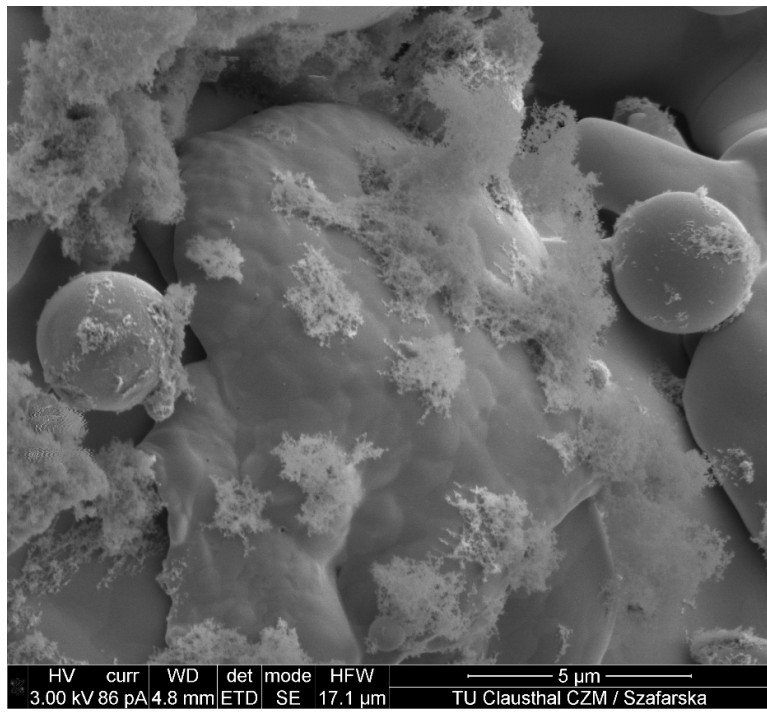

**Figure 16.** SEM micrograph of a copper coating surface with undeformed microparticles and fine $SiO_2$ agglomerates.

It can be assumed that the silicon dioxide is not only deposited on the surface of coating, but is also present at the interface between the substrate and the coating, and is also embedded within in the coating. To what extent this influences the bond strength is the subject of ongoing research.

## 5. Conclusions

Wire arc spraying of copper in an environment with an oxygen partial pressure corresponding to the extreme-high vacuum (XHV) was studied with a focus on porosity and adhesive tensile strength. The main results can be summarized as follows:

1.  The coatings formed in under XHV-adequate conditions are free of oxide seams, feature few interfacial gaps, and have a significantly reduced porosity.
2.  By transferring the blasting process to the oxygen-free environment, existing oxide films can be removed and their new formation suppressed. This improves the wetting behaviour of the impacting particles.
3.  The oxide-free interfaces result in substantial increased adhesive tensile strength. In some cases, the strength exceeded 60 MPa, which corresponds to an 154% increase as compared to the data obtained for conventionally sprayed coatings.
4.  The inert spay atmosphere, i.e., absence of oxygen, has a strong influence on the particle and splat formation, which can be attributed to the associated change in surface tension.

**Author Contributions:** Conceptualization, M.R.D., M.S., R.G., K.M. and H.J.M.; methodology, M.R.D.; software, M.R.D. and M.S.; validation, M.R.D. and M.S.; formal analysis, M.R.D. and M.S.; investigation, M.R.D. and M.S.; resources, R.G., K.M. and H.J.M.; data curation, M.R.D. and M.S.; writing—original draft preparation, M.R.D.; writing—review and editing, M.R.D., M.S., R.G., K.M. and H.J.M.; visualization, M.R.D. and M.S.; supervision, R.G., K.M. and H.J.M.; project administration, R.G., K.M. and H.J.M.; funding acquisition, R.G., K.M. and H.J.M. All authors have read and agreed to the published version of the manuscript.

**Funding:** Funded by the Deutsche Forschungsgemeinschaft (DFG, German Research Founda-tion)—Project-ID 394,563,137—SFB 1368.

**Institutional Review Board Statement:** Not applicable.

**Informed Consent Statement:** Not applicable.

**Data Availability Statement:** The data presented in this study are available upon request from the corresponding author.

**Acknowledgments:** This project was funded by the Deutsche Forschungsgemeinschaft (DFG, German Research Foundation)—Project-ID 394,563,137—SFB 1368.

**Conflicts of Interest:** The authors declare no conflict of interest.

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
