# Peer review of "Oxide Free Wire Arc Sprayed Coatings—An Avenue to Enhanced Adhesive Tensile Strength"

_metals, doi:10.3390/met12040684_

Round 1

Reviewer 1 Report

This article sounds good however some parts are missing. Such as at beginning article type. 

Figure 2 with and without blasting gun curves are both bent. I could see same response from green line. How do author explain this?

A table should include to represent sample coated using and types of sample. In this way it will be clear idea about samples, parameter and differences. 

Figure 6 need to be more clear. more descriptive ways. Figure 8 need to be more clear with sub head a, b and c. 

Some related article need to include

Fabrication of thermal plasma sprayed NiTi coatings possessing functional properties

  Coatings 11 (5), 610, 2021.   Net-Shape NiTi Shape Memory Alloy by Spark Plasma Sintering Method   Appl. Sci. 11 (4), 1802,2021

Reviewer 2 Report

The paper presents the results regarding the arc-spraying of the metallic coatings. The authors investigated the effect of spray atmosphere (inert / air) on the microstructure and adhesion of the coatings. 

Paper presents interesting results and generally is well written and structured. I have comments regarding the research material naming and overall suggestions pointing out that surface roughness and EDS investigations should be added.

  • you wrote in MDPI system that this is a review paper while it should be changed to "research paper"
  • please use SI units - change "bar" to MPa
  • It is advisable to number the important features in the photos, in fig. 1.
  • It is unclear while you show the III, IV and V (three columns for 
    blasted and coated in silane + N2
    Please clarify it in the caption.
  • The same in the text "
    Specimen sets (III), (IV), and (V) 203
    were all blasted and coated in silane-doped nitrogen.

    " - explain it in the text.
  • Moreover, in the fig. 12 specimens codes V, Vi and VII are presented. Please clarify in methodology section the specimens codes and clearly write what are the differences between different sets of specimens.
  • I think that it is necessary to mention the roughness (Ra, Sa, Rz or St parameters) of the sprayed coatings. Measure it and discuss the roughness. The roughness of as-sprayed coatings affects the adhesion test results. Roughness results can be given in "morphology section"
  • I suggest authors use SEM-EDS to clearly state the microstructure phases. You mention oxidation in your paper. The oxidation of both types of specimens should be investigated. mark the oxidised phases in the microstructure cross-section - EDS mapping is advisable. 
  • EDS will support your conclusions - you wrote "oxide-free" and "absence of oxygen". By the way, the last conclusion is inappropriate. You should clearly state that" intert spray process atmosphere i.e. absence of oxygen " rather than "absence of oxygen".

Round 2

Reviewer 2 Report

Thank you for your responses. I accept your explanations and have no further comments on your paper.